# Design of synthetic epigenetic circuits featuring memory effects and reversible switching based on DNA methylation

Johannes A.H. Maier[1], Raphael Möhrle[1] & Albert Jeltsch[1]

Epigenetic systems store information in DNA methylation patterns in a durable but reversible manner, but have not been regularly used in synthetic biology. Here, we designed synthetic epigenetic memory systems using DNA methylation sensitive engineered zinc finger proteins to repress a memory operon comprising the CcrM methyltransferase and a reporter. Triggering by heat, nutrients, ultraviolet irradiation or DNA damaging compounds induces CcrM expression and DNA methylation. In the induced on-state, methylation in the operator of the memory operon prevents zinc finger protein binding leading to positive feedback and permanent activation. Using an *mf*-Lon protease degradable CcrM variant enables reversible switching. Epigenetic memory systems have numerous potential applications in synthetic biology, including life biosensors, death switches or induction systems for industrial protein production. The large variety of bacterial DNA methyltransferases potentially allows for massive multiplexing of signal storage and logical operations depending on more than one input signal.

[1] Institute of Biochemistry, Faculty of Chemistry, Stuttgart University, Stuttgart 70569, Germany. Correspondence and requests for materials should be addressed to A.J. (email: albert.jeltsch@ibc.uni-stuttgart.de).

The design of artificial, DNA encoded regulation circuits is a key goal of synthetic biology, and in 2000 two seminal papers described genetic circuits realizing a toggle switch and an oscillator[1,2]. Since then, various circuit design principles have been developed and tested in bacterial or mammalian cell systems. Examples are circuits for cell population regulation[3], an epigenetic transgene switch in mammalian cells[4], systems for light detection by *Escherichia coli*[5], genetic memory systems based on recombinases[6,7], a bistable genetic switch[8], examples of circuits performing basic Boolean logic operations and memory effects[9], design of bacteria invading cancer cells[10] and kill switches useful for biocontainment[11].

Here we aimed to apply an epigenetic mechanism based on DNA methylation for artificial gene regulation in *E. coli*. Epigenetic signals are defined as being heritable but also reversible providing an option to design circuits with stable switching and memory effect, but still being resettable. In eukaryotic cells, epigenetic modifications lead to cell differentiation and include DNA methylation and hydroxymethylation, histone posttranslational modifications and non-coding RNAs[12]. In bacteria, epigenetic effects are mediated by DNA methylation, mainly adenine-N6 methylation, which controls the interaction of DNA binding proteins and DNA interacting enzymes with their target sites, and is implicated in many important biological processes like host defence against bacteriophage infection, post-replicative mismatch repair or control of the initiation of DNA replication[13–15]. Switchlike behaviours of bacterial epigenetic systems have been described among others for the paradigmatic *agn43* gene, which is involved in biofilm formation and for pyelonephritis-associated pili expression in uropathogenic *E. coli*[16,17].

Combining synthetic circuit design with epigenetic mechanisms, we constructed different synthetic epigenetic memory systems in *E. coli* allowing to sense temporary stimuli and memorize this information for many cell generations. The bistable memory systems described here consist of artificial regulatory networks, which store information in form of DNA methylation patterns in a reversible manner. Inspired by reports describing engineered zinc finger (ZnF) proteins and TALE repressors that are sensitive to cytosine-C5 methylation[18,19], we engineered an artificial adenine-N6 methylation dependent DNA binding protein using the ZnF chassis. Employing this ZnF protein and the CcrM methyltransferase from *Caulobacter crescentus*, which introduces DNA-(adenine N6)-methylation at GANTC sites[20,21], we constructed a bistable basic module for epigenetic memory systems, which can exist in two states, an on-state and an off-state. In the initial off-state, the ZnF repressor binds to the promoter region of the *ccrM* gene and represses its expression (Fig. 1a). Once the binding of the ZnF repressor is weakened, CcrM is expressed and can methylate its own promoter region. DNA methylation of the ZnF binding site further weakens ZnF binding resulting in a positive feedback loop. Coupling different trigger systems to this memory device, we were able to detect and memorize physical properties like temperature or ultraviolet radiation and the presence of soluble metabolites or toxic agents.

## Results

**Design concept of the epigenetic memory system.** Using a multi-step design process, we set up an epigenetic memory system in *E. coli* aiming to use orthologous components that do not interfere with *E. coli* physiology. Key components of our system are a DNA methyltransferase (MTase), a ZnF protein that binds to DNA in a methylation sensitive manner and serves as a repressor, coiled-coil elements for controlled protein oligomerization and a targeted protein degradation system. As MTase part, we have chosen the CcrM DNA-(adenine N6)-methyltransferase from *Caulobacter crescentus* that methylates adenine residues in GANTC sequences. The second part of our system, a ZnF protein that does not bind to CcrM methylated DNA was designed by us. Taking natural bacterial repressors as a paradigm, we multimerized the ZnF protein to increase the DNA binding strength and used autoregulation to achieve a stable and low-expression of the ZnF protein. To set up a memory system, we designed a promoter controlled by CcrM methylation to regulate the expression of a reporter–maintenance operon including an EGFP reporter and the CcrM MTase, which after expression methylates its own promoter establishing a positive feedback. Next, we used different sensor elements and coupled them to the newly developed epigenetic system to make it responding to temperature changes, soluble metabolites and genotoxic stress.

**Effects of CcrM expression on *E. coli* viability.** One big challenge in synthetic biology is the development of parts that do not interfere with the physiology of the host organism, *E. coli* in our case. To investigate if CcrM expression affects the viability of *E. coli*, CcrM was induced using a pBAD-CcrM vector by addition of arabinose and the growth rates of the expression cells were measured after 3 and 5 h of induction. As control, experiments were conducted with an inactive CcrM mutant carrying a D31A mutation in the highly conserved DPPY motif of the enzyme. Mutations of this aspartate to alanine have been shown to inactivate DNA-(adenine N6)-MTases[22]. Moreover, expression of the unrelated NusA protein was tested and, as positive control, the CpG specific DNA-(cytosine-C5)-MTase M.SssI was used, which is known to be toxic in *E. coli*[23]. As shown in Supplementary Fig. 1a, we observed that CcrM expression did not affect the growth rate of *E. coli* over 3 h under our experimental conditions, and it only caused a minor reduction in growth rate after 5 h. *E. coli* cells expressing catalytically inactive CcrM or NusA showed similar small growth reductions, indicating that the effect is not due to CcrM activity, but likely caused by the general overexpression conditions of the pBAD system. In contrast, expression of M.SssI led to a rapid and strong reduction of cell proliferation. The lack of adverse growth effects of GANTC methylation in *E. coli* is in agreement with the fact that < 4% of all *E. coli* transcription factors have such a sequence in their core binding motif (Supplementary Fig. 1b).

**ZnF protein screening for methylation dependent DNA binding.** To build an artificial bacterial repressor based on a ZnF protein that can be modulated by DNA-(adenine N6)-methylation, we started designing C2H2 zinc finger arrays recognizing DNA sequences including or overlapping with GANTC sites that can be methylated by CcrM. The design of methylation specific DNA binding was based on the classical recognition of an AT base pair by Asn or Gln with two hydrogen bonds to N6 and N7 of the adenine, which would be disrupted by N6 methylation (Supplementary Fig. 2a). ZnF protein building blocks binding the desired DNA sequences were selected using Zinc Finger Targeter (Supplementary Fig. 2b)[24,25]. We then used a bacterial two-hybrid reporter assay[26], to test five designed ZnF proteins for binding of their target sequences. In this assay, a *lacZ* reporter gene is used that has the ZnF binding site in its promoter region. Expression of the reporter is driven by the recruitment of an RNA polymerase-Gal4 fusion protein to target promoters by a ZnF-Gal11p fusion protein such that ß-Gal activity is only observed, if the ZnF protein binds to its target sequence. Using this approach, we detected DNA binding with 4 of the ZnF

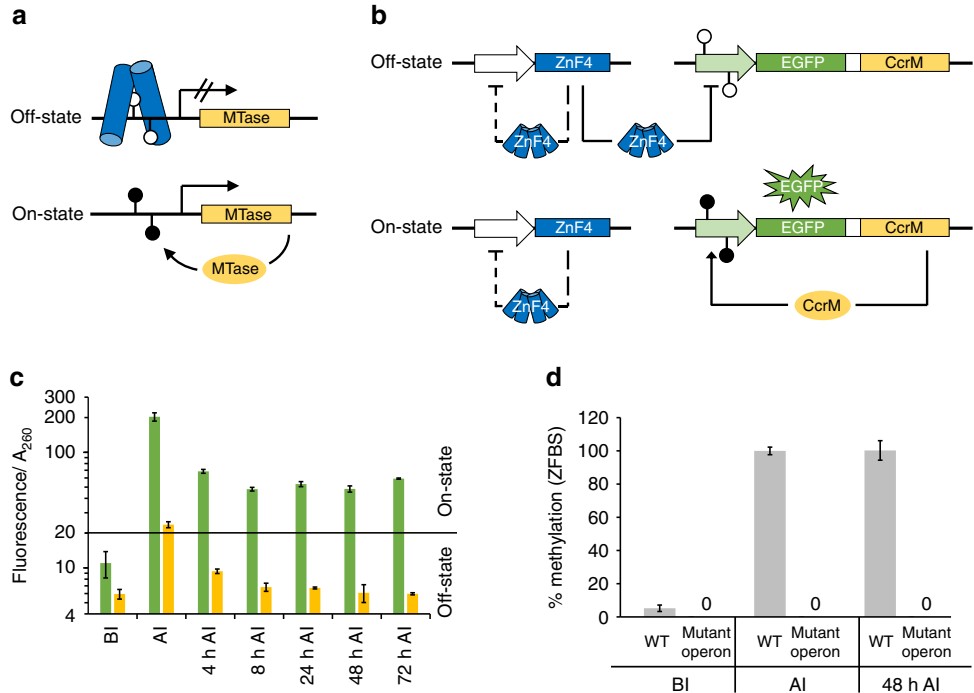

**Figure 1 | Heat induction of memory system I. (a)** Scheme of the synthetic epigenetic memory system I. In the off-state, the synthetic ZnF repressor binds the promoter region of a methyltransferase (MTase) gene. In the on-state, the promoter region is methylated, repressor binding is hindered and the MTase gene is transcribed resulting in a positive feedback loop. Filled and open lollipops represent methylated and unmethylated GANTC sites. **(b)** Circuit design of the synthetic epigenetic memory system with ZnF repressor and reporter–maintenance operon. In the off-state, ZnF4 inhibits transcription of the reporter-maintenance operon containing EGFP and CcrM. Once the system is switched to the on-state, EGFP and CcrM are expressed and the binding of the ZnF repressor is prevented by DNA methylation. CcrM constantly re-methylates the operator ZnF binding sites and keeps the system in the on-state. The ZnF repressor regulates its own expression in a manner not affected by methylation (indicated by negative feedback with dashed lines). **(c)** Fluorescence data of XL1-Blue cells transformed with the memory plasmid with reporter–maintenance operon. Cells were cultivated at 30 °C in the off-state (BI, before induction) followed by overnight induction at 37 °C (AI, after induction). Further cultivation of the cells was done at 30 °C and fluorescence measured 4–72 h after switching back to 30 °C (green bars). Orange bars represent the control experiment with a mutant reporter–maintenance operon containing a catalytically inactive CcrM variant (error bars indicate s.d., $n = 3$ biological repeats). The vertical line separates the signal ranges characteristic for the on- and off-state. **(d)** Restriction-protection-qPCR analysis of promoter methylation of the reporter-maintenance operon (see Supplementary Fig. 12 for details of the assay). For the reporter–maintenance operon with active CcrM (WT), the operator was unmethylated in the off-state (BI). The operator was methylated after heat induction and methylation was maintained for at least 48 h. The operator of the control CcrM mutant reporter–maintenance operon was unmethylated throughout the experiment (error bars indicate s.d., $n = 3$ biological repeats). BI: before induction, AI: after overnight induction by cultivation at 37 °C, AI 48 h: after overnight induction and 48 h cultivation at 30 °C. For statistics, see Supplementary Table 1.

proteins (Supplementary Fig. 2b). In the second step, we tested DNA binding after co-expression with CcrM to analyse the impact of adenine methylation in the ZnF binding site. Binding to methylated DNA was reduced for all of the ZnF proteins, albeit to a different degree (Supplementary Figs 2 and 3). We identified ZnF_1012 as the best candidate for further applications, as it showed strong binding compared to the no-zinc-finger-protein control, but no detectable binding in the presence of CcrM. During the course of the project, we fused the ZnF protein to coiled-coil domains for dimerization (ZnF2)[27] and tetramerization (ZnF4)[28], as multimerization of DNA binding proteins is a common feature in nature to increase binding affinity[29,30]. We tested methylation dependent DNA binding of ZnF4 using an electromobility shift assays (EMSA) assay *in vitro* and observed a roughly 70-fold preference for the unmethylated DNA binding site compared to the methylated DNA binding site (Supplementary Figs 4 and 5).

**Construction of a methylation sensitive repressor system.** We constructed an artificial gene expression system utilizing the zinc finger protein (ZnF_1012), as a repressor. Using the pET28a(+) plasmid as an initial vector scaffold, we removed all T7 regulatory elements and placed a *gfp* reporter gene under the control of a

constitutive promoter. Two ZnF_1012 binding sites were inserted enclosing the −35 promoter region. We replaced the gene coding for the lac repressor by the gene coding for ZnF_1012 (Supplementary Fig. 6a), but observed that ZnF_1012 did not efficiently repress GFP expression (Supplementary Fig. 6b). We then fused the coiled-coil domain from GCN4 for dimerization[27] to ZnF_1012, resulting in ZnF2, and observed strong repression of the *gfp* gene (Supplementary Fig. 6b). Afterwards, we tested the methylation sensitivity of ZnF binding to the promoter region by using the second plasmid encoding an inducible CcrM (Supplementary Fig. 7a) revealing that GFP expression was only observed after CcrM induction, but not under CcrM repressing conditions (Supplementary Fig. 6c).

**Construction of epigenetic memory system I.** Next, we aimed to construct an epigenetic memory system, which uses the methylation sensitive repressor to control the expression of a reporter–maintenance operon encoding an EGFP reporter and the CcrM MTase that can methylate the repressor binding site in its own promoter (Fig. 1). This system was expected to show strong positive feedback allowing to memorize transient signal detection. To set up a memory system, a very strong repression of the reporter–maintenance operon in the off-state was necessary to

avoid spontaneous activation due to the positive feedback. Taking natural repressor systems as examples, we therefore fused the synthetic zinc finger protein ZnF_1012 to the coiled-coil domain from GCN4-p-LI (ref. 28) for tetramerization yielding ZnF4, which can simultaneously bind to two palindromic double recognition sites, both regulated by DNA methylation. We introduced three palindromic double binding sites for the ZnF repressor upstream and downstream of the transcriptional start site (Supplementary Fig. 8). The first double binding site was designed to surround the $-35$ region of the promoter as in the synthetic methylation sensitive repressor system described above. The second double binding site was placed 100 bps upstream of the first one in a manner resembling the arrangement of pseudo-operator sites in the lac operon[31]. The third double binding site was placed between the *egfp* reporter gene and maintenance *ccrM* gene to prevent spontaneous transcription initiation upstream of the *ccrM* gene (Supplementary Fig. 9). All these binding sites can be methylated by CcrM to regulate the repressor binding. Moreover, we introduced an additional double binding site for the ZnF repressor in the promoter region of its own gene in order to maintain a low steady-state level of repressor molecules by autoregulation (Supplementary Figs 8 and 10). This autoregulatory binding site was designed not to overlap with CcrM sites, hence, it cannot be methylated and repressor binding is independent of CcrM expression. In the initial off-state, the ZnF4 repressor can bind all unmethylated binding sites (Fig. 1b), and as expected almost no EGFP expression was observed indicating tight repression (Fig. 1c).

**Physical induction of memory system I**. To sense a physical parameter, we turned our attention to temperature and showed that an overnight shift of the cultivation temperature of the *E. coli* cells from 30 to 37 °C weakens the binding of the ZnF repressor to its binding sites resulting in a derepression of the *egfp* and *ccrM* genes. Expression of CcrM leads to methylation of the ZnF binding sites and this in turn prevents the binding of the ZnF repressor installing a positive feedback, which leads to stable switching of the system into the on-state (Fig. 1c). Control experiments demonstrated that stable repression was only achieved in the presence of all three ZnF double binding sites (Supplementary Fig. 9), and stable on-state switching was only possible with autoregulation of ZnF4 expression (Supplementary Fig. 10).

As a control to investigate the dependence of the reporter–maintenance operon on CcrM for stable activation, we used the CcrM D31A mutant. Systems containing the reporter–maintenance operon encoding the catalytically inactive CcrM mutant only revealed a slight increase in fluorescence immediately after heat induction (Fig. 1c). Already 4 h later, EGFP levels were reduced below the on-state threshold. In contrast, with the system containing the active CcrM in the reporter–maintenance operon, EGFP expression was maintained when the cells were further cultivated at 30 °C for at least 72 h, corresponding to $\sim 36$ generations (based on the experimentally observed doubling time of these cells at 30 °C of about 2 h) (Fig. 1c). This finding indicates that the constant re-methylation of the ZnF binding sites is critical for maintaining the on-state, as methylation is constantly lost passively by DNA replication. Additional experiments showed that the on-state was even stable after re-streaking of bacterial cells on agar-plates to single colonies and re-inoculation of liquid cultures (Supplementary Fig. 11a and b).

We analysed the methylation state of the promoter region of the memory system by methylation dependent restriction digestion with the restriction endonuclease HinfI followed by quantification of undigested DNA by qPCR (Supplementary Fig. 12). In the initial off-state, we observed very low levels of methylation (5.2% ± 1.9%) with the CcrM wildtype reporter–maintenance operon (Fig. 1d). After heat induction, almost complete methylation was observed, which stayed high even after cultivation at 30 °C for 48 h (Fig. 1d). In the control samples containing the inactive CcrM mutant in the reporter–maintenance operon, no methylation was observed (Fig. 1d).

**Construction and chemical induction of memory system II**. Next, we aimed to expand the versatility of the system by using a chemical trigger for switching the epigenetic memory system, in this case arabinose, representing a metabolite and nutrient of *E. coli*. For this, we used a second plasmid encoding a tightly regulated *ccrM* gene (now called trigger CcrM) that can be induced by arabinose. For distinction of on-states based on acute induction or memory effects, we included the gene for the red fluorescent protein mCherry in the trigger operon (Fig. 2a, Supplementary Fig. 7b and 13). As expected, a drastic increase of the red fluorescence signal was observed after addition of arabinose, indicating the successful activation of the trigger operon. Identical red fluorescence signals were observed, no matter if the memory system contained an active or inactive CcrM (now called maintenance CcrM), indicating that the maintenance CcrM plays no role in this initial response (Fig. 2b). However, after removing arabinose from the medium, the red fluorescent signals disappeared rapidly, while EGFP fluorescence was maintained for at least 96 h (Fig. 2c) indicating that the memory system has undergone a stable switching from the off- to the on-state. As expected, no stable switching into the on-state was observed with an inactive maintenance CcrM (Fig. 2c), showing that the presence of an active maintenance CcrM enzyme is required for positive feedback and system switching. On-switching of memory system II occurred after overnight incubation with at least 0.01% arabinose (Supplementary Fig. 14). Stable switching was also achieved after 4 h incubation with 0.2% arabinose, while cultivation with 0.2% arabinose for only 2 h led to a delayed switching phenotype (Supplementary Fig. 15). Additional experiments indicated that the on-state was stable even throughout competent cells generation and electroporation of *E. coli* cells (Supplementary Fig. 11c and d).

Cultivation experiments showed no difference in growth rates of bacteria in the on- or off-state of memory system II (Supplementary Fig. 16). Flow cytometry analysis revealed homogeneous cell populations in the off- and the on-states, although the EGFP fluorescence levels varied considerably among cells, documenting noise in the gene expression in both states (Fig. 2d). The EGFP intensities of single cells in the off- and on-state were also easily distinguished by laser scanning confocal microscopy (Fig. 2e). Statistical analysis of fluorescence intensities of individual cells revealed a normal distribution in the on-state and hence confirmed results acquired by flow cytometry (Fig. 2f).

**Construction and induction of memory system III**. Next, we were interested to explore the potential of epigenetic switches with memory function to detect a transient exposition of bacteria to toxins or ultraviolet irradiation and store this information in the on-state of the memory system in form of DNA methylation. We decided to construct a DNA damage sensor element, because the DNA damage SOS response is a general response to ultraviolet radiation, radioactive exposition and exposition to chemical mutagens. The trigger plasmid was constructed using a modified version of a promoter from the *E. coli* plasmid pColD-157 (refs 32–34) to control the trigger MTase expression (Supplementary Fig. 17). The ColD promoter is repressed by LexA and only active under SOS response conditions when RecA initiates self-cleavage

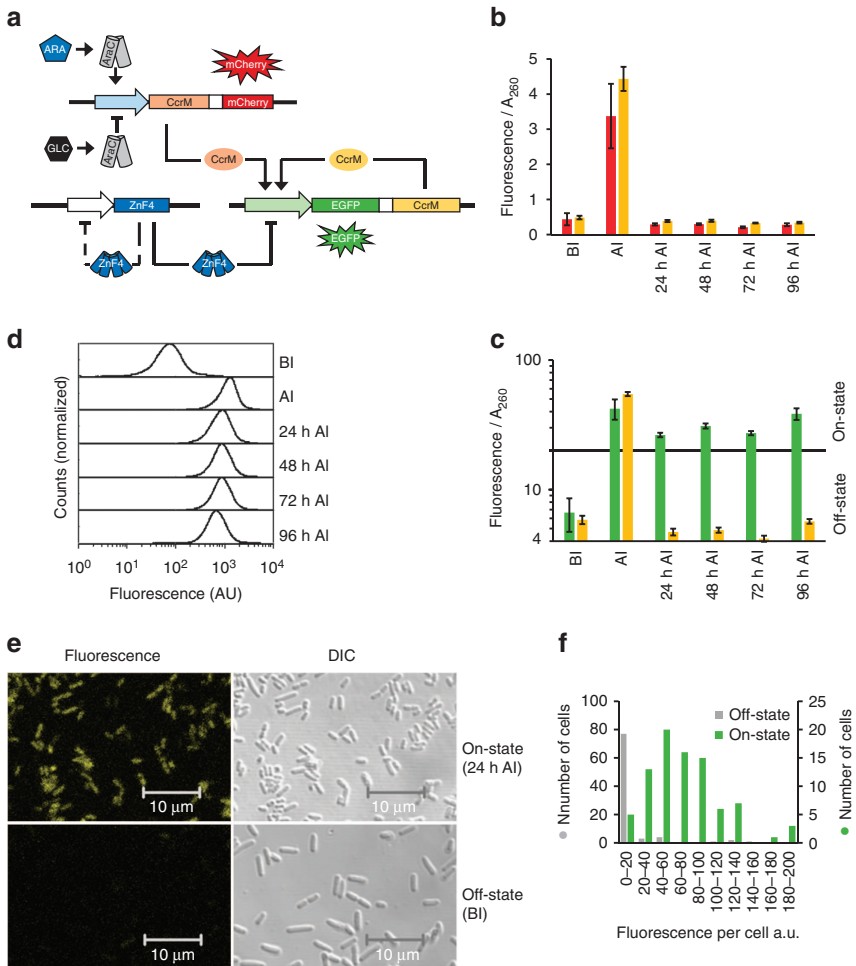

**Figure 2 | Chemical induction of memory system II by arabinose.** (**a**) Circuit design of memory system II. In the initial off-state, glucose represses the trigger CcrM expression. The on-state is induced by arabinose supplementation, which induces CcrM and mCherry expression from the trigger plasmid. After removal of arabinose, the on-state is maintained by expression of the reporter–maintenance operon via a positive feedback loop. (**b**) Red fluorescent protein (mCherry) fluorescence measured in total cell lysate. Red bars: wildtype CcrM in the reporter–maintenance operon; orange bars: active site mutant CcrM in the reporter–maintenance operon (BI: before induction, that is, in glucose containing media; AI: after induction, that is, cultivation for 12 h in presence of arabinose; 24 h AI, 48 h AI, 72 h AI, 96 h AI, that is, cultivation in glucose containing media for the indicated time period after cultivation for 12 h in arabinose containing media) (error bars indicate s.d., $n = 3$ biological repeats). mCherry signal can only be detected in the presence of arabinose. (**c**) EGFP signal measured in total cell lysate. On induction with arabinose, green fluorescence levels rise above the threshold level and stay in the on-state range for at least 96 h after induction (green bars). If the catalytically inactive CcrM is present in the reporter–maintenance operon, EGFP fluorescence disappears 24 h after induction and no memory function can be observed (orange bars) (error bars indicate s.d., $n = 3$ biological repeats). (**d**) EGFP signal measured by flow cytometry. Cells with functional reporter–maintenance operon (that is, wildtype CcrM) were analysed. Histograms depict homogeneous populations in the off- and on-state. 20,000 events were collected for each measurement. (**e**) Confocal laser scanning microscopy pictures and differential interference contrast (DIC) microscopy pictures of cells with functional reporter–maintenance operon in the on-state (24 h AI) and cells in the off-state (BI). (**f**) Quantification of fluorescence intensities of cells in the on-state and in the off-state recorded by confocal laser scanning microscopy ($n = 87$, per state). For statistical analysis of the results shown in **b**,**c**, see Supplementary Table 2.

of LexA[35]. This leads to de-repression of the promoter, transcription of the trigger MTase and subsequently to the switching of the memory system to the on-state (Fig. 3a; Supplementary Fig. 18). Transient SOS response was initiated by ultraviolet light exposition or cisplatin addition to the medium. Cells co-transformed with the trigger plasmid for sensing DNA damage and the memory plasmid showed an increase in EGFP levels after induction of DNA damage either by ultraviolet or cisplatin (Fig. 3b and c). We measured a stable on-state for at least 48 h in cells with functional reporter–maintenance operon. Cells transformed with the same SOS response trigger plasmid but a reporter–maintenance operon containing the inactive CcrM also showed increased EGFP signals after induction, albeit with lower overall intensity (Fig. 3b and c).

However, in these cells EGFP levels dropped to background levels after stopping of ultraviolet or cisplatin treatment.

**Construction of memory system IV for signal reset.** One key advantage of epigenetic systems is that they are reversible, since no genetic changes are involved. We therefore aimed to introduce options for signal termination and enforced return of the system to the off-state after activation. For this, we used a synthetic protein degradation system[36] to selectively degrade the maintenance CcrM and stop further DNA methylation. The Lon protease from *Mesoplasma florum* (*mf*-Lon) is orthologous to the *E. coli* Lon, because it degrades proteins with *M. florum* ssrA tag, but not *E. coli* ssrA tagged proteins, and, *vice versa*, *E. coli* Lon does not degrade

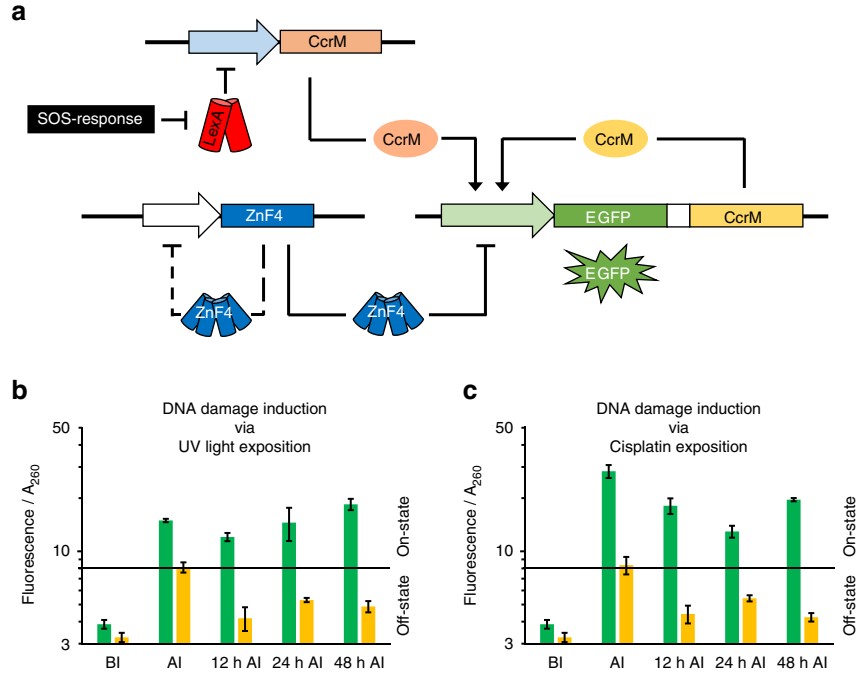

**Figure 3 | Induction of memory system III by DNA damage. (a)** Circuit design of the DNA damage sensor with memory function. The trigger CcrM MTase is negatively regulated by the LexA repressor. On DNA damage, *E. coli* induces the SOS-response, which leads to cleavage of the LexA repressor and subsequently to the expression of the trigger MTase. **(b)** EGFP signal of ultraviolet-irradiated cells ($3 \times 50 \, \mathrm{J \, m^{-2}}$ energy intake). The on-state signal is maintained by the reporter-maintenance operon for at least 48 h (error bars indicate s.d., $n = 2$ biological repeats). **(c)** EGFP signal of cells treated with cisplatin. Cisplatin was removed from the medium after 12 h. Cells stayed in the on-state for at least 48 h (error bars indicate s.d., $n = 2$ biological repeats). For a more detailed representation, see Supplementary Figs 17 and 18. For statistical analysis of the results shown in **b,c**, see Supplementary Table 3.

*M. florum* ssrA tagged proteins[37]. We C-terminally fused the protein degradation tag pdt#2 (ref. 36) to the maintenance CcrM resulting in CcrM-deg, and cloned an arabinose inducible *mf-lon* gene into the second plasmid (Fig. 4a; Supplementary Fig. 19a).

Cells co-transformed with plasmids coding for inducible *mf*-Lon, and the heat inducible memory system containing CcrM-deg were cultured at 30 °C in the off-state. We switched on the memory system by heat induction and subsequently were able to switch it to the off-state again by arabinose induction of the expression of *mf*-Lon, which selectively degrades CcrM-deg. Altogether, we showed that the system could be switched on and off in two cycles (Fig. 4b; Supplementary Fig. 19b). As a control, we used cells harbouring the same plasmids, but containing an untagged version of the *ccrM* gene. No sensitivity to *mf*-Lon expression could be observed and cells stayed in the on-state for the whole culturing time, after the initial heat induction (Fig. 4b; Supplementary Fig. 19c). Stability of the on-state of the irreversible and reversible system was confirmed by culturing on-state cells without induction of *mf*-Lon (Supplementary Fig. 20), similarly as shown before.

## Discussion

Our data illustrate that it is possible to store information in DNA methylation patterns in dividing bacteria throughout many cell divisions if appropriate positive feedback systems are established. We demonstrate this by constructing artificial epigenetic circuits in *E. coli* that can process various transient signal inputs (physical signals like heat or ultraviolet light and chemical signals like toxins or nutrients) and memorize the appearance of transient signals in form of DNA methylation at GANTC sites. Combining orthologous and synthetic genetic parts, this synthetic epigenetic memory circuit can function autonomously in *E. coli* (sensing heat or arabinose) or as an accessory system attached to

endogenous circuits (sensing SOS response). We show here that the methylation signal encoding the on-state of the system was stable over many cell generations and during physiological changes including the generation of competent cells and transformation of bacteria. The epigenetic memory systems described here are based on a designed ZnF protein that does not bind adenine-N6 methylated DNA. To further extend the toolbox for constructing similar systems, designed natural DNA binding proteins or TAL effectors could be applied as well. Moreover, measurements of the MTase, ZnF and reporter protein and mRNA concentrations, cell growth and DNA methylation rates in the different states of the system will allow quantitative modelling of the system's behaviour and support future design projects.

Potential applications of the systems described here could be in the area of biocontainment, after introducing a gene for a toxin into the memory system, which is stably induced after a certain signal is provided. This could be used to kill genetically modified bacteria that have escaped from a bioreactor, but also to terminate a vaccination with live bacteria in case of medical complications. The synthetic epigenetic memory system could also be used in the context of large scale industrial protein expression, where induction of regulated expression systems is costly and sometimes difficult to control. In the systems developed here, inducers like IPTG would be required only transiently to activate the expression of the gene of interest. Other applications are in the field of live bacterial biosensors. Utilizing the SOS trigger combined with the memory system, a biosensor for radioactivity or toxins could be developed, for example in form of small encapsulated and buffered biosensor spheres containing live bacteria. Similar biosensors with triggers responding to different toxins or pollutants are easily conceivable, which would allow a cost efficient, long-term observation of environmental sites for industrial pollution or contamination. Controlling the

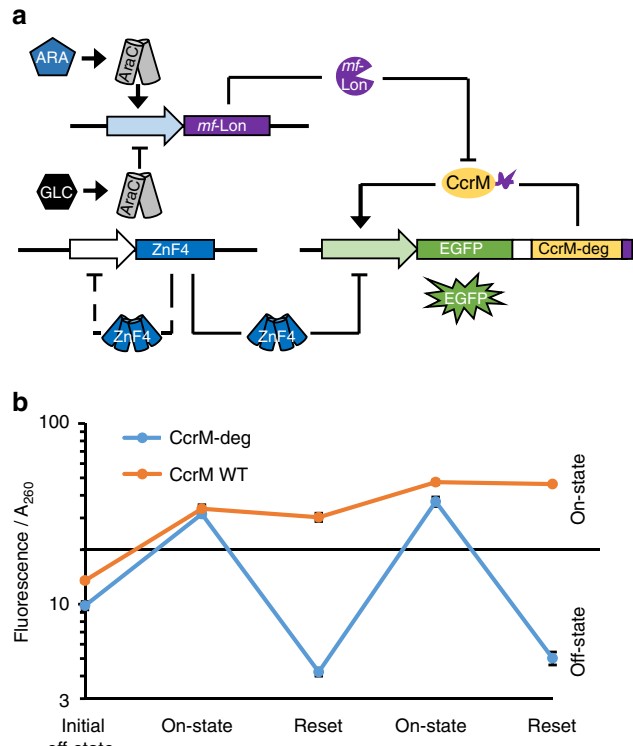

**Figure 4 | Signal reset of memory system IV.** (**a**) Circuit design of the memory system including an inducible *mf*-Lon protease and a CcrM carrying a protein degradation tag (CcrM-deg), which leads to its specific degradation by *mf*-Lon. Afterwards, adenine-N6 methylation at GANTC sites is passively lost and the system returns to the off-state. (**b**) Experimental results of cycles of switching the memory system to the on-state by temperature induction and its return to the off-state by induced CcrM degradation (blue trace). The original system I containing wildtype CcrM without protein degradation tag did not respond to *mf*-Lon expression (orange trace). The figure shows stable steady-state signals in the corresponding phases for three individually cultivated clones (error bars indicate s.d., $n = 3$ biological repeats). For a more detailed representation and additional control experiments, see Supplementary Figs 19 and 20.

maintenance of cooled supply chains for pharmaceutical drugs, chemicals or agricultural products could be another area of application. In the field of biomedical diagnosis, it is conceivable to develop sensor systems that respond to biomarkers of diseases, for example, tumour markers in the gut. Biomedical diagnosis systems could be administered as biosensor pills that travel through a patient and collect their corresponding signal input for readout after excretion similar to a described genetic tetracycline sensor[38].

One particular advantage of epigenetic systems is that resetting of the sensor is possible, for example, immediately before use or after one successful signalling event. This is relevant, because systems with positive feedback loops tend to show long-term instability caused by stochastic on-switching. In future, the advantages and disadvantages of epigenetic and genetic switch systems with respect to sensitivity, stability and reversibility need to be determined and compared in the light of the requirements of the different applications.

The system developed by us stores information of transient input signals in the form of DNA-(adenine N6)-methylation patterns in bacterial cells. Of note, there are hundreds of characterized bacterial adenine-N6 MTases with different target sequences[39], and many different DNA methyltransferases can co-exist in individual bacterial strains. Hence, in principle, it should be possible to assemble several independent memory systems in one bacterial cell, which could collect several different inputs and store them in form of different DNA methylation patterns in parallel, which are set by individual MTases and read by specifically designed DNA binding proteins. Promising candidates of additional MTases are the EcoRI (GAATTC) and EcoRV (GATATC) enzymes, which naturally occur in *E. coli* isolates and can be recombinantely expressed in lab strains[40,41], indicating that they do not cause strong physiological responses in *E. coli*, similar as CcrM used here. Combinations of the described systems with different methylation marks like cytosine-C5 methylation are also conceivable. Single-molecule real-time sequencing provides a cost efficient method for readout of complex DNA methylation patterns[42,43] that is no longer dependent on fluorescent proteins or other reporter proteins. Finally, the combination of different input signals with different DNA methylation patterns would also allow to connect them and construct logic gates to perform initial steps of data analysis and integration.

## Methods

**Molecular cloning and bacterial cell culturing.** All cloning procedures were performed by Gibson assembly unless stated otherwise[44]. *E. coli* XL1-Blue cells were used for the design of all synthetic epigenetic systems. Cells were cultured in a way that they never exceeded $OD_{600\,nm}$ of 1.

**Zinc finger design and methylation dependent DNA binding.** We were aiming to design zinc finger proteins that bind to target sequences overlapping with GANTC sites in unmethylated state, but that do not bind GANTC methylated target sites. To this end, five different target DNA sites overlapping with GANTC motifs were selected. Then, three domain C2H2 ZnF proteins binding to these sequences were identified by Zinc Finger Targeter from the Zinc Finger Database[25,45] and ZnF arrays were tested for DNA binding in *E. coli* using a bacterial two-hybrid system as described[26] (Addgene Kit # 1000000010). The sensitivity of ZnF binding to methylated target sites was tested by co-expression of the system with CcrM. Thereto, the gene coding for CcrM together with a ribosomal binding site (5′-TTT GTT TAA CTT TAA GAA GGAG A-3′) was cloned into the pAC-Kan-alphaGal4 plasmid downstream of the kanamycin resistance gene creating a polycistronic operon. Expression of the *ccrM* gene and its activity was tested by digestion of plasmid DNA isolated from these bacteria with HinfI, which is an adenine methylation sensitive restriction enzyme cleaving GANTC sites (Supplementary Fig. 3). DNA binding of the synthetic ZnF fusion proteins could be detected for all tested variants but for ZnF_604 (Supplementary Fig. 2). After DNA methylation with CcrM, all ZnF fusion proteins showed reduced DNA binding. We chose to investigate ZnF_1012 in further experiments, as it showed strong DNA binding to its unmethylated target site and no detectable DNA binding to its methylated target site (Supplementary Fig. 2).

**Cloning and purification of MBP-tagged ZnF_1012.** To study the DNA binding of ZnF_1012 *in vitro*, we expressed and purified the protein. The gene for ZnF_1012 fused to the GCN4-p-LI coiled-coil (ZnF4) was cloned into the pMAL-c2X (NEB) expression vector resulting in a maltose-binding protein (MBP) fusion construct. *E. coli* BL21-CodonPlus (DE3) cells were used for protein over-expression. Cells transformed with pMal-c2x-ZnF4 were grown in Luria broth (LB) medium supplemented with 25 μg ml$^{-1}$ kanamycin and 10 μM $ZnSO_4$. Protein expression was induced in the mid logarithmic growth phase ($OD_{600\,nm} \approx 0.5$) at 30 °C by addition of 0.3 mM IPTG. Cells were collected after 4 h of induction by centrifugation and stored at −20 °C until further use. Cells were lysed by sonication in 50 mM Tris–HCl pH 8.5, 500 mM NaCl, 5% glycerol, 0.2 mM DTT and 1 mM PMSF. Lysates were cleared by centrifugation for 1 h at 18,000g. The cleared lysate was applied to amylose resin (#e8021s NEB) in a gravity flow column at 8 °C. Extensive washing was performed with lysis buffer without PMSF. Protein was eluted with washing buffer supplemented with 20 mM maltosemonohydrate. Purified protein was dialysed against a buffer containing 50 mM Tris–HCl, pH 8.5, 200 mM NaCl, 5% glycerol and 0.2 mM DTT. Concentration of the purified protein was measured spectroscopically using an extinction coefficient of 69,330 M$^{-1}$ cm$^{-1}$. Protein was frozen in aliquots in liquid nitrogen and stored at −80 °C. Protein purity was assessed by SDS–polyacrylamide gel electrophoresis (SDS–PAGE) (Supplementary Fig. 4d).

***In vitro* ZnF protein DNA binding.** DNA binding of purified MBP-tagged ZnF4 was assayed via EMSA using different Cy-5 labelled 152 bp PCR fragments as substrates, one that did not contain the ZnF binding sites, and one with 2 ZnF binding sites, which were used in unmethylated and methylated form (Supplementary Fig. 4a and b). The methylated DNA substrate was prepared by

*in vitro* DNA methylation of the unmethylated substrate with purified His$_6$-tagged CcrM[46]. Methylation of the PCR product (60 ng μl$^{-1}$) was performed in reaction buffer (50 mM HEPES pH 7.0, 50 mM NaCl, 1 mM EDTA, 0.5 mM DTT, and 5 μg ml$^{-1}$ BSA) supplemented with 200 μM S-adenosyl-L-methionine (Sigma) and 5 μM CcrM for 1 h at room temperature. DNA was purified afterwards with NucleoSpin Gel and PCR Clean-up kit (Macherey-Nagel). DNA methylation was confirmed by digestion with the methylation sensitive restriction endonuclease HinfI (NEB) (Supplementary Fig. 4c). For the DNA binding studies, 5 nM Cy-5 labelled DNA was incubated with 5 nM MBP-ZnF4 in binding buffer (5 mM Tris–HCl, pH 8.5, 210 mM NaCl, 4.8% glycerol, 0.05% Tween 20, 8 mg l$^{-1}$ BSA) for 30 min in the dark at room temperature. Samples were run on a 5% non-denaturating polyacrylamide gel in 0.5 × Tris–borate buffer. Cy-5 fluorescence signal was detected with a FUSION Solo (Peqlab) system and signal intensities were quantified with ImageJ[47]. Different unlabelled DNA molecules were added, as competitors in increasing concentrations to probe the specificity of the binding reaction.

**Assembly of a methylation dependent gene expression system.** The pET28(a) + plasmid (Novagen) served as an initial vector scaffold for the assembly of a synthetic methylation dependent gene expression system. The region ranging from the T7 promoter to the T7 terminator was removed and replaced with a GFP expression cassette consisting of a constitutive *E. coli* sigma70 promoter (BBa_J23107), a ribosomal binding site (BBa_B0032), a gene coding for GFP and a terminator (BBa_B0010). Two ZnF binding sites were placed in palindromic orientation in the promoter region with a distance of 5 bp, the second of which was partially overlapping with the − 35 region. The GFP expression cassette was obtained as a synthetic DNA sequence from Eurofins Genomics (Germany) and cloned into the pET28(a) + plasmid. The gene for LacI was replaced by a gene coding for FLAG-tagged ZnF_1012 and in a second construct fused to a coiled-coil motif for dimerization (GCN4, PDB ID: 2ZTA (ref. 27), (ZnF2). Cloning of the coiled-coil motif was performed via Gibson assembly of oligonucleotides[48]. To increase the expression levels of ZnF constructs, the lacI promoter was altered to lacIQ1 (ref. 49) (Supplementary Fig. 6a and b). For analysis of methylation sensitive repression of the *gfp* reporter gene, the methyltransferase CcrM was cloned into the pBAD24 vector[50] (Supplementary Fig. 7a). XL1-Blue electro competent cells were co-transformed with the GFP reporter plasmid and CcrM encoding pBAD24 vector, and plated on LB agar plates containing kanamycin (25 μg ml$^{-1}$) and ampicillin (100 μg ml$^{-1}$). Expression of the *ccrM* gene in liquid culture was induced by addition of arabinose (0.2% (w/v)).

**Construction of the synthetic epigenetic memory system.** Starting from the synthetic methylation dependent gene expression system, a synthetic epigenetic memory system was created. *gfp* was replaced by *egfp* and the gene for CcrM together with a ribosome binding site (BBA_B0064) was cloned downstream of the *egfp* reporter gene resulting in an artificial operon (reporter-maintenance operon). Two additional palindromic DNA double binding sites for the zinc finger repressor protein with overlapping CcrM sites were introduced into the system (100 bp upstream of the already existing binding site and 8 bp downstream of the ribosome binding site for the *ccrM* gene). One palindromic double ZnF binding site that does not overlap with GANTC sites and hence was not methylated by CcrM was introduced next to the − 35 region of the zinc finger repressor promoter for autoregulation (Supplementary Figs 8 and 9). The zinc finger protein was fused to a GCN4-p-LI domain, which mediates the formation of a homotetramer (PDB ID: 1GCL)[28]. For the annotated full sequence of the memory system, see Supplementary Note 1.

**Fluorescence measurements.** For fluorescence measurements, cells were collected by centrifugation and resuspended in phosphate buffer containing PopCulture Reagent (Merck Millipore). OD$_{260 nm}$ was measured and adjusted to 1.5. All fluorescent measurements were normalized to OD$_{260 nm}$. GFP fluorescence measurements were performed with HitachiF-2500 Fluorescence Spectrophotometer. Excitation wavelength was set to 495 nm and fluorescence was recorded in the range from 505 to 525 nm with a scan speed of 300 nm min$^{-1}$. Excitation and emission bandwidth were set to 2.5 nm. Autofluorescence of *E. coli* XL1- Blue cells not containing any fluorophore was subtracted. EGFP and mCherry fluorescence measurements were performed with a Jasco FP-8300 Fluorescence Spectrofluorometer. For EGFP fluorescence measurements, excitation wavelength was set to 488 nm and fluorescence was recorded in the range from 495 to 525 nm with a scan speed of 200 nm min$^{-1}$. Excitation and emission bandwidth were set to 2.5 nm. High sensitivity settings were used and the response time was set so 0.2 s. For mCherry fluorescence measurements, excitation wavelength was set to 587 nm and fluorescence was recorded in the range from 592 to 650 nm with a scan speed of 200 nm min$^{-1}$. Excitation and emission bandwidth were set to 2.5 nm. High sensitivity settings were used and the response time was set so 0.2 s. For bar representation of fluorescence measurements, the mean values of fluorescence intensities in the interval of ± 2 nm of the emission maximum (EGFP 509 nm, mCherry 610 nm) were calculated. All fluorescence measurements were performed from at least two separately treated and cultured samples.

**Methylation state analysis.** The methylation state of the ZnF binding sites in the promoter region of the reporter–maintenance operon was assessed by quantitative

PCR. Plasmid DNA was isolated from 1 ml bacterial liquid culture using NucleoSpin Plasmid (Macherey-Nagel) and digested with restriction endonucleases EcoRI-HF and HinfI in buffer provided by the supplier (NEB) followed by heat inactivation at 80 °C. 5 pg of the digested plasmid DNA was used as template for quantitative PCR. The promoter region of the memory system was amplified using specific primers designed using Primer3 software[51,52] (FP: 5′-ACA CCC GTA AAC AGC TCC TC-3′, RP: 5′-CGG CGT AGA GGA TCG AGA T-3′). The amplicon includes two GANTC sites that can be methylated by CcrM. In the unmethylated state, the GANTC sites are cleaved by the restriction endonuclease HinfI. When methylated, the sites are protected from HinfI digestion. As a reference for plasmid input, a part of the kanamycin resistance gene not containing a HinfI restriction site was amplified (FP: 5′-AAT GGG CTC GCG ATA ATG TC-3′, RP: 5′-CAA CGC TAC CTT TGC CAT GT-3′). For each sample biological and technical triplicates were used for methylation analysis. S.d. were calculated from biological replicates. qPCR experiments were conducted with a CFX96 Real-Time system (Bio-Rad) using SsoFast EvaGreen supermix (Bio-Rad). Standard curves for each primer pair were recorded with undigested template DNA to assess the PCR efficiency.

**Trigger plasmid responding to arabinose induction.** Starting from the pBAD24 (ref. 50) plasmid containing the *ccrM* gene (Supplementary Fig. 7a), the pBR322 origin of replication was replaced by the p15A origin of replication from pACYC184 plasmid[53]. In addition, the gene for the red fluorescent protein mCherry[54] was cloned directly downstream of the *ccrM* gene separated by a ribosome binding site (BBa_B0034) for direct fluorescence reporting of active transcription on arabinose induction (Supplementary Fig. 7b).

**Flow cytometry.** Cells were collected by centrifugation and resuspended in PBS with Ca$^{2+}$ and Mg$^{2+}$ to an OD$_{600 nm}$ of ∼0.2. Approximately 20,000 events were acquired using a BD FACS Calibur flow cytometer at low-flow rate and EGFP signal was recorded. Data were analysed using FCS Express V4 (De Novo Software).

**Fluorescence microscopy imaging.** Microscopy glass slides (Thermo Scientific, ER-240B-CE24, 10 wells, 6 mm) were polylysinated (0.1% (w/v) Poly-L-lysine in H$_2$O (Sigma)), 10 μl of bacterial culture was added and covered with a coverslip. Cells were imaged on an LSM 710 Zeiss confocal microscope with a Plan-Apochromat × 63/1.40 Oil DIC M27 objective directly after slide preparation. Fluorescence intensities of individual cells were quantified with ImageJ[47].

**Trigger plasmid responding to DNA damage.** Starting from the pBAD24 vector carrying the *Caulobacter crescentus ccrM* gene (Supplementary Fig. 7a), the P$_{BAD}$ promoter together with the *araC* gene was removed and an SOS responsive promoter was inserted instead. To this end, a modified version of the promoter region from the *cda* gene from the *E. coli* plasmid pColD-157 (Genbank: Y10412.1) was used to regulate trigger CcrM expression[32,33]. In addition, wildtype *recA* gene was cloned downstream of ampicillin resistance gene together with spacer and ribosomal binding site (BBa_B0035) (5′-AGG TCG TTT CTA GAG ATT AAA GAG GAG AAT ACT AG-3′). The origin of replication of the plasmid was changed to the p15A origin of replication from pACYC184 plasmid[53] (Supplementary Fig. 17).

**SOS response induction by ultraviolet light and cisplatin.** Electro competent *E. coli* XL1-Blue cells were co-transformed with the memory system plasmid and SOS response trigger plasmid. Cells were cultured for 1 h at 30 °C in SOC medium supplemented with 30 μM ZnSO$_4$. Cells were then plated on LB agar plates containing 25 μg ml$^{-1}$ kanamycin, 100 μg ml$^{-1}$ ampicillin, 30 μM ZnSO$_4$ and 0.2% glucose and incubated at 30 °C for 24 h. Single colonies were used to inoculate overnight cultures in LB medium containing 25 μg ml$^{-1}$ kanamycin and 100 μg ml$^{-1}$ ampicillin, 10 μM ZnSO$_4$ and 0.2% glucose. Cells were treated with ultraviolet light in order to induce SOS response. A 5 ml sample of a liquid bacterial culture was transferred to a petri dish (9 cm diameter) and ultraviolet radiation was applied with a UVC 500 Ultraviolet Crosslinker (Amersham) using 50 J m$^{-2}$ of ultraviolet energy per surface area. Treatment was repeated three times with 2 h breaks for recovery. To induce SOS response via DNA damaging chemicals, cells were treated with 50 μM of cisplatin (Acros organics) for 12 h. After treatment with cisplatin cells were washed two times with medium not containing cisplatin.

**Cloning of *mf-lon* protease and protein degradation tag fusion.** The *mf-lon* gene (Addgene plasmid #21867)[37] was cloned into pBAD24 containing the p15A origin of replication. The memory plasmid containing the reporter-maintenance operon was modified by the addition of protein degradation tag pdt#2 (ref. 36) to CcrM resulting in CcrM-deg.

**Data availability.** All relevant data supporting the findings of this study are available from the authors on reasonable request.

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

## Acknowledgements

We gratefully acknowledge C. Lungu for help with microscopy imaging, Dr G. Kungulovski for help with qPCR experiments and Dr P. Rathert for help with flow cytometry.

## Author contributions

J.A.H.M. and A.J. designed the study. J.A.H.M. contributed to all experiments. R.M. contributed to cloning and purification of zinc finger protein, EMSA experiments and reset switch. J.A.H.M. and A.J. analysed and interpreted the data. J.A.H.M. and A.J. wrote the manuscript draft. All authors approved the final version of the manuscript.
