## [Peer Review File · Nature Communications]

Reviewers' Comments:

Reviewer #1 (Remarks to the Author)

This manuscript describes a proof-of-concept for the design of DNA methylation-based epigenetic sensors. The idea is brilliant and has few precedents in the literature. The description of the sensors and the analysis of their workings may be too succinct to evaluate their actual performance, but the underlying ideas are relevant enough in synthetic biology to consider this manuscript interesting for a major journal. The following questions and comments may help the authors to nail down some critical points and to improve the writing.

1. General comment. The sensors described in this manuscript are plasmid-borne. The stability of the plasmids, alone and when used in pairs, should be monitored.
2. General comment. How many GATC sites exist in the *E. coli* genome? Is it correct to say that introduction of the heterologous DNA methylase CcrM does not interfere with *E. coli* physiology? Have the authors tested whether introduction of CcrM affects the growth rate?
3. Introduction, page 1. *agn43* and *pap* are classical examples of DNA methylation-dependent switches but there are others (*pef*, *opvAB*, etc.). I suggest that the sentence is modified indicating this point. It may suffice to indicate that *agn43* and *pap* are classical examples and that others exist.
4. Results, page 3. Do you mean "orthogonal" components?
5. Results, page 4. This comment may be nitpicky. I suggest use of the term "nonmethylation" instead of "unmethylation" as the latter is better suited to describe active, Tet-dependent removal of C5 methylation marks in eukaryotes.
6. Results, page 4. Is eGFP the optimal choice among the existing GFPs? Is it stable enough to enable memory? This point may be relevant to evaluate the sensitivity of the sensors.
7. Results, page 6. Typo in "re-inoculation".
8. Results, page 7. Flow cytometry does not reflect the absence of bacterial subpopulations but expression is quite noisy. This point should be mentioned in the text.

Reviewer #2 (Remarks to the Author)

The paper concerns the use of DNA methylation patterns as a recording device for temporary perturbations such as heat, nutrient availability, UV irradiation etc. in *E. coli*. The bistable system described in this manuscript consists of a synthetic regulatory network that stores information in form of DNA methylation in a reversible manner. The work is novel because as the authors state it is the first time that epigenetics and DNA methylation has been used as a basis for a synthetic regulatory gene circuit in *E. coli*.

This paper will be of interest to researchers engaged in the Synthetic Biology field. It is also an interesting tool for bacteriologists.

The work is convincing, appears thorough and the conclusions justified.

A few minor comments and things to address:

Page 1 Abstract

"Inclusion of a protein degradation tag allowed to establish a reversible switching system". This sentence doesn't make sense. Allowed us?

Results Section

Page 4 para 1 Some clarification needed on the premise that the ZnF protein should not bind to

the methylated DNA recognition site is required.

The description of the ZnF protein design is not very clear, lacks detail and is difficult to follow. The way that it is written in the methods section is much better but there is a lack of crucial detail in the results on what the different designed proteins consist of and how exactly they differ.

State how the DNA binding assays were done – no details but say which technique.

Supplementary figure 4 is poor and a better image is required to illustrate conclusions.

Figure 1 Need to state what AI and BI stand for in the figure legend. Figure 1.C need to explain how the on- or off- state is being represented. Not sure that 1.D is necessary. Figure 1.E - it is unclear from the figure and legend what the AI section is illustrating. What time frame compared to the AI 48h?

Reviewer #3 (Remarks to the Author)

In my opinion the manuscript "Design of synthetic epigenetic circuits exhibiting positive feedback, memory effects and reversible switching" in its current form is too limited to meet the standard required for publication in this journal. The concept of employing methylation markers in a synthetic gene circuit is novel and interesting, and the experimental data convincingly prove the successful operation of the epigenetic switch in the contexts that were tested. However, the data were not detailed enough to provide useful design criteria for other researchers wishing to use these components. In addition, the real impact of this study would come from the demonstration of the broader utility of this design, which remains unproven.

The manuscript describes the development of a bistable genetic switch based on a DNA methyltransferase that regulates its own expression via modification of the operator region of its promoter; methylation blocks binding of an engineered zinc-finger repressor, setting expression of the methyltransferase (and reporter) to a stable ON state that is stably maintained over multiple generations. The circuit could be switched on in response to heat due to unstable binding of the repressor at higher temperatures, and in response to arabinose and DNA damage when a trigger element consisting of the methyltransferase with inducible promoter was included. Finally, reversible switching using an inducibly degradable version of the methyltransferase was demonstrated. The core functionality of all of these circuits was demonstrated clearly.

One area of the switch development that was unclear was the necessity of zinc-finger protein multimerisation using coiled-coil domains to improve ZnF binding to the operator. There is no comparison of ZnF binding vs ZnF2 vs ZnF4, nor a description of how DNA binding sites were designed to accommodate the multimeric versions. These are important design considerations for this system. The ZnF/ZnF2/ZnF4 terminology was also sometimes confused.

In general there is a lack of detail about the behaviour of the switch in vivo. How fast does switching (ON and OFF) occur? What are the effective ranges of inducer strength and induction period? How do the fluorescence readings compare to a standard? This lack of detail makes it difficult to grasp how the system compares to other genetic tools. If the system were characterised in greater depth, using the data to address the following questions would strengthen the manuscript:

- What advantages/disadvantages does this design of bistable switch have over existing switches? (In terms of switching speed, stability, robustness...)
- What burden does this system place on the cell? How does this burden scale?
- How can the switch characteristics be tuned?
- Do the characteristics of this switch class (relative and absolute ON/OFF states, switching kinetics...) allow for easy integration into larger circuits?

Supplementary figure 14 is essential proof that the reversible system is stable, but is not referred to in the main text.

The methods section is generally sufficiently detailed and clear, but could be condensed slightly; some details are repeated in the results section, and some results creep into the methods. However, a lack of information also exists in places, for example, what method was used to maintain cell cultures at $OD_{600} < 1$? A chemostat? Also, I am unfamiliar with the reasoning behind normalising cell-extracts using the absorbance at 260 nm – could this be explained? Why was it not possible to take fluorescence readings from intact cells? It would be good to see annotated whole plasmid sequences uploaded to a public repository.

The English is generally very good, but the manuscript could be enhanced by proofreading by a native English speaker.

Regarding the issue of broader applicability of the design, the authors note that there are many characterised methyltransferases with different specificities, and that multiple methyltransferase systems could operate in parallel to record signals or create logic functions. However, there are many potential pitfalls to the expansion of this system that ought to be acknowledged and explored. Ultimately a demonstration of multiple engineered epigenetic switches operating in parallel is required.

The choice of methyltransferase enzyme needs to be justified. A short recognition motif (only four specific bases for the CcrM enzyme used here) will be present at fairly high frequency in a bacterial genome. The authors state that the CcrM enzyme has no reported effects in *E. coli*, but this is an important consideration that needs validation, especially when considering larger systems with multiple methyltransferases. There is no simple test of orthogonality performed in the present work, for example a comparison of growth rates between wild-type and methyltransferase-expressing cells. A better validation would be a comparison of RNA-seq data from those strains. Can the authors justify why they did not investigate a system which couples methyltransferase activity to a more specific DNA binding protein, for example a zinc-finger protein or deactivated-Cas9? The programmability of other DNA binding proteins might widen the applicability of the authors' epigenetic switch principle.

Likewise, are zinc-finger proteins the best class of repressor to use? There is no justification given. Would TALE repressors, which are more easily programmed to specific sequences, be better? What other classes of repressor are responsive to methylation of their binding site, and could the system be used to interface with gene circuits built from other classes of components?

The choice of mechanism to reverse the switch also limits the expansion of this class. For example, using small RNAs to inhibit translation of the methyltransferase would allow specific enzymes to be knocked down when many are operating in parallel in a gene circuit.

Whilst the use of DNA methylation to build a gene circuit is novel, it has been used for other bio-engineering applications. Additional references to reflect this background to the present work would be beneficial. Other references to consider in general:

Repurposing the CRISPR-Cas9 system for targeted DNA methylation; Vojta et al; NAR; 2016
Reprogrammable CRISPR/Cas9-based system for inducing site-specific DNA methylation; McDonald et al; Biol. Open; 2016

Specific targeting of cytosine methylation to DNA sequences in vivo; Smith and Ford; NAR; 2007
Recognition of methylated DNA by TAL effectors; Deng et al; Cell Res.; 2012

Engineered zinc finger proteins that respond to DNA modification by HaeIII and HhaI methyltransferase enzymes; Isalan and Choo; JMB; 2000.

Chimeric DNA methyltransferases target DNA methylation to specific DNA sequences and repress expression of target genes; Li et al; NAR; 2007

Synthetic epigenetics—towards intelligent control of epigenetic states and cell identity; Jurkowski et al; Clin. Epigenetics; 2015

Design of synthetic epigenetic circuits exhibiting positive feedback, memory effects and reversible switching

NCOMMS-16-26741

Reply to the reviewers' comments

General comments

We appreciate the insightful and constructive comments of all reviewers which have helped us to significantly improve the manuscript. Based on them, new experiments were conducted, which are now shown in several panels of Supplementary Figures as detailed below. In addition, we have improved writing at several places. Due to this and further rewriting, it was necessary to reorder the Supplementary Figures. To aid the comparison of the revised version with the original one, we provide a list of the numbering of the Supplementary figures in both versions below.

Revised version: Suppl. Fig. number and panel	Original version
1	New figure
2A	New panel
2B	1
3	2
4	3
5	4
6	5
7	6
8	7
8D	New panel
9	New figure
10	8
10C	New panel
11	10
12	Article Figure 1 D
13	11
14	New figure
15	New figure
16	New figure
17	12
18	13
19	14
20	15
20C	14C
Supplementary Text 1	New information

Please note that in this document new Supplementary File designations are used throughout.

Reply to the comments of reviewer 1 (printed in blue)

This manuscript describes a proof-of-concept for the design of DNA methylation-based epigenetic sensors. The idea is brilliant and has few precedents in the literature. The description of the sensors and the analysis of their workings may be too succinct to evaluate their actual performance, but the underlying ideas are relevant enough in synthetic biology to consider this manuscript interesting for a major journal. The following questions and comments may help the authors to nail down some critical points and to improve the writing.

Reply: Thanks a lot for this positive assessment and your helpful comments.

1. General comment. The sensors described in this manuscript are plasmid-borne. The stability of the plasmids, alone and when used in pairs, should be monitored.

Reply: Throughout the work, when plasmids were used in combination, they contained different origins of replication and different antibiotic resistance genes. The memory operon containing plasmids exhibit pBR322/ColE1 origin of replication and a kanamycin resistance gene, the sensor plasmids carry p15A origin of replication and an ampicillin resistance gene. Compatibility of p15A and pBR322/ColE1 origins of replications is well documented in literature (Chang and Cohen 1978, Journal of bacteriology, PMID: PMC222365). During all cultivation steps, appropriate antibiotics have been present, preventing potential loss of plasmids. Hence, no problems regarding plasmid stability were expected. There was one exception to this in one of the intermediate design step (shown in Supplementary Fig. 6C). Here, two different plasmids with the same origin of replication (pBR322/ColE1) were used, however, the stable presence of both plasmids in cells was ensured by individual antibiotic resistance genes on each plasmid and appropriate antibiotics in the culturing medium.

2. General comment. How many GATC sites exist in the *E. coli* genome? Is it correct to say that introduction of the heterologous DNA methylase CcrM does not interfere with *E. coli* physiology? Have the authors tested whether introduction of CcrM affects the growth rate?

Reply: The *E. coli* K12 genome (*Escherichia coli* K-12 substr. MG1655, GenBank: U00096.3) comprises 4.6 million bps containing about 10,750 GATC sites. Based on our previous experience in recombinant expression of CcrM in *E. coli*, we assumed that CcrM has no drastic influence on *E. coli* physiology. For confirmation, we analyzed the growth rates of *E. coli* (XL1-Blue) cultures transformed with pBAD24_CcrM under expression conditions and repressed conditions. In addition, we used PBAD24_M.SssI, which is known for some toxic effects in *E. coli*, for comparison under the same conditions. We included the results of these experiments in the Supplementary information of the article (Supplementary Fig. 1).

Additionally we checked growth rates in the on-state and in the off-state of XL1 Blue cells carrying the memory system II and could not detect strong differences in the growth rates in the different states (Supplementary Fig. 16), indicating that switching does not influence viability of *E. coli*.

To further accommodate the concern behind this comment, we changed the writing the main text now saying: “we set up an epigenetic memory system in *E. coli* aiming to use orthogonal components that do not interfere with *E. coli* physiology”.

3. Introduction, page 1. *agn43* and *pap* are classical examples of DNA methylation-dependent switches but there are others (*pef*, *opvAB*, etc.). I suggest that the sentence is modified indicating this point. It may suffice to indicate that *agn43* and *pap* are classical examples and that others exist.

Reply: We have changed the writing according to the suggestion.

4. Results, page 3. Do you mean "orthogonal" components?

Reply: We have changed the writing according to the suggestion.

5. Results, page 4. This comment may be nitpicky. I suggest use of the term "nonmethylation" instead of "unmethylation" as the latter is better suited to describe active, Tet-dependent removal of C5 methylation marks in eukaryotes.

Reply: We understand and appreciate the logic behind using "nonmethylated" for sites generated by passive demethylation and "unmethylated" for sites generated by active demethylation. However, in our simplistic system this distinction is not necessary, because there is no active demethylation. Also, this nomenclature has not been widely adopted and many authors use both terms as synonyms. We have inspected the usage of both terms with google revealing 623000 occurrences of "unmethylated" but only 162000 of "nonmethylated". We would, hence, propose to keep the more widely used term, but we are willing to change this upon discretion of the editor.

6. Results, page 4. Is eGFP the optimal choice among the existing GFPs? Is it stable enough to enable memory? This point may be relevant to evaluate the sensitivity of the sensors.

Reply: Please note that the system provides a continuous expression of the eGFP gene. The protein stability of eGFP is not involved in the memory function.

7. Results, page 6. Typo in "re-inoculation".

Reply: Thank you for mentioning, we corrected the mistake.

8. Results, page 7. Flow cytometry does reflect the absence of bacterial subpopulations but expression is quite noisy. This point should be mentioned in the text.

Reply: We included this information in the text.

Reply to the comments of reviewer 2 (printed in blue)

The paper concerns the use of DNA methylation patterns as a recording device for temporary perturbations such as heat, nutrient availability, UV irradiation etc. in *E. coli*. The bistable system described in this manuscript consists of a synthetic regulatory network that stores information in form of DNA methylation in a reversible manner. The work is novel because as the authors state it is the first time that epigenetics and DNA methylation has been used as a basis for a synthetic regulatory gene circuit in *E. coli*.

This paper will be of interest to researchers engaged in the Synthetic Biology field. It is also an interesting tool for bacteriologists.

The work is convincing, appears thorough and the conclusions justified.

A few minor comments and things to address:

Reply: Thank you very much for this positive assessment and the helpful comments.

Page 1 Abstract “Inclusion of a protein degradation tag allowed to establish a reversible switching system”. This sentence doesn’t make sense. Allowed us?

Reply: The sentence has been rewritten: “Reversible switching could be achieved using a CcrM variant that can be degraded by an inducible mf-Lon protease. “

Results Section Page 4 para 1 Some clarification needed on the premise that the ZnF protein should not bind to the methylated DNA recognition site is required.

Reply: We have included one additional schematic panel in Supplementary Fig. 2A, showing the principal idea of the design, i.e. that adenine methylation will disrupt the recognition of an AT base pair by Gln or Asn.

The description of the ZnF protein design is not very clear, lacks detail and is difficult to follow. The way that it is written in the methods section is much better but there is a lack of crucial detail in the results on what the different designed proteins consist of and how exactly they differ.

Reply: We have rewritten the beginning of the results section for clarity. We also included more detailed information of the ZnF protein design in Supplementary Fig. 2, including the new panel mentioned above. We provide the annotated sequences of the regulatory elements with ZnF binding sites indicated (Supplementary Fig. 8D). We also added data showing that multimerization of the ZnF protein is necessary for repressor function (Supplementary Fig. 6B).

State how the DNA binding assays were done – no details but say which technique.

Reply: We have included more description in the corresponding section, describing the principle of the bacterial two hybrid assay and mentioning that in vitro DNA binding was analyzed by EMSA.

Supplementary figure 4 is poor and a better image is required to illustrate conclusions.

Reply: We now show a better image with optimized contrast settings that allows to visualize the gel shift better. We also expanded the description of the analysis and show a new image panel explaining the analysis (Supplementary Fig. 5B).

Figure 1 Need to state what AI and BI stand for in the figure legend. Figure 1.C need to explain how the on- or off- state is being represented. Not sure that 1.D is necessary. Figure 1.E - it is unclear from the figure and legend what the AI section is illustrating. What time frame compared to the AI 48h?

Reply: We included descriptions of AI and BI in the figure legends and expanded the legend of Fig. 1C. We moved Figure 1D to the supplementary part as requested.

Reply to the comments of reviewer 3 (printed in blue)

One area of the switch development that was unclear was the necessity of zinc-finger protein multimerisation using coiled-coil domains to improve ZnF binding to the operator. There is no comparison of ZnF binding vs ZnF2 vs ZnF4, nor a description of how DNA binding sites were designed to accommodate the multimeric versions. These are important design considerations for this system. The ZnF/ZnF2/ZnF4 terminology was also sometimes confused.

Reply: We added to Supplementary Fig. 6 additional data showing that ZnF without multimerization domain is insufficient to repress gene expression in the intermediate design stage of the repression system. Moreover, we added Supplementary Fig. 9 showing that all three palindromic double ZnF binding sites are needed for stable off-state maintenance of the memory system. We added Supplementary Fig. 10C, showing that autoregulation was needed for stable on-state maintenance of the memory system. We also added an additional figure panel in Supplementary Fig. 8 (Supplementary Fig. 8D) that shows location and orientation of ZnF binding sites. We changed writing in the results part of the main article, now mentioning the need for multimerization of the ZnF repressor for functionality. We corrected wrong ZnF designations in the legend of Supplementary Fig. 4 and in the method section.

In general there is a lack of detail about the behaviour of the switch *in vivo*. How fast does switching (ON and OFF) occur? What are the effective ranges of inducer strength and induction period? How do the fluorescence readings compare to a standard? This lack of detail makes it difficult to grasp how the system compares to other genetic tools.

Reply: We did additional experiments to characterize the system further. We used memory system II to investigate necessary inducer concentrations to switch the system to the on-state by overnight induction (Supplementary Fig. 14). Additionally, we checked for how long the induction signal (arabinose) has to be present in order to switch the system to the on-state (Supplementary Fig. 15).

If the system were characterised in greater depth, using the data to address the following questions would strengthen the manuscript:

- What advantages/disadvantages does this design of bistable switch have over existing switches? (In terms of switching speed, stability, robustness...)

Reply: In this study, we do not conduct comparison to existing switches; we rather want to prove the feasibility of developing synthetic epigenetic switches in bacteria based on DNA-(adenine N6)-methylation.

- What burden does this system place on the cell? How does this burden scale?

- How can the switch characteristics be tuned?

- Do the characteristics of this switch class (relative and absolute ON/OFF states, switching kinetics...) allow for easy integration into larger circuits?

Reply: We like to mention, that our study was the first example showing this kind of bacterial epigenetic system. We agree with the reviewer, that depending on the later application, more detailed experiments need to be conducted. To provide some answers to the questions raised, we

tested the influence of CcrM expression on *E. coli* physiology by analyzing the growth rates of *E. coli* (XL1-Blue) cultures transformed with pBAD24_CcrM under expression conditions and repressed conditions. In addition, we used PBAD24_M.Sssl, which is known for some toxic effects in *E. coli*, for comparison under the same conditions. We included the results of these experiments in the Supplementary information of the article (Supplementary Fig. 1). Moreover, we determined growth rates in the on-state and in the off-state of XL1-Blue cells carrying the memory system II and could not detect strong differences in the growth rates in the different states (Supplementary Fig. 16), indicating that switching does not strongly influence viability of *E. coli*.

Supplementary figure 14 is essential proof that the reversible system is stable, but is not referred to in the main text.

Reply: Thank you for this hint. We added a sentence to the main manuscript stating that the control experiments document that the memory system is stable, similarly as shown before.

The methods section is generally sufficiently detailed and clear, but could be condensed slightly; some details are repeated in the results section, and some results creep into the methods. However, a lack of information also exists in places, for example, what method was used to maintain cell cultures at $OD_{600} < 1$? A chemostat? Also, I am unfamiliar with the reasoning behind normalising cell-extracts using the absorbance at 260 nm – could this be explained? Why was it not possible to take fluorescence readings from intact cells? It would be good to see annotated whole plasmid sequences uploaded to a public repository.

Reply: Thank you for these hints. We have focused on adding of missing information and now specified that cells were grown in test cultures and diluted manually with medium, in order to keep $OD_{600} < 1$. We measured the fluorescence intensities in cell lysates in order to reduce light scattering by whole cells and this resulted in a dramatic improvement of signal to noise. As shown for memory system II it is also possible to see the fluorescence signal in intact cells depicted in the microscopy pictures and FACS analysis. In order to normalize fluorescence signals to cellular amount, we used absorbance at 260 nm. Alternatively, normalization to A280 would have been also possible, but since the protein A280 peak is in the shoulder of the A260 peaks, in our hands A260 resulted in better reproducibility. We added the annotated plasmid sequences of memory System I as a supplement (Supplementary Text 1).

The English is generally very good, but the manuscript could be enhanced by proofreading by a native English speaker.

Reply: Thanks a lot. We have critically read the manuscript once more and corrected additional language mistakes. We hope that writing is now clear in all parts.

Regarding the issue of broader applicability of the design, the authors note that there are many characterised methyltransferases with different specificities, and that multiple methyltransferase systems could operate in parallel to record signals or create logic functions. However, there are many potential pitfalls to the expansion of this system that ought to be acknowledged and explored. Ultimately a demonstration of multiple engineered epigenetic switches operating in parallel is required.

Reply: We agree with the reviewer, but setting up a second orthogonal system is beyond the scope of the current work. In nature, many different DNA methyltransferases co-exist in bacteria, illustrating that these enzymes often are compatible with each other. This point has been added to the discussion section.

The choice of methyltransferase enzyme needs to be justified. A short recognition motif (only four specific bases for the CcrM enzyme used here) will be present at fairly high frequency in a bacterial genome. The authors state that the CcrM enzyme has no reported effects in *E. coli*, but this is an important consideration that needs validation, especially when considering larger systems with multiple methyltransferases. There is no simple test of orthogonality performed in the present work, for example a comparison of growth rates between wild-type and methyltransferase-expressing cells. A better validation would be a comparison of RNA-seq data from those strains.

Reply: Based on our previous experience in recombinant expression of CcrM in *E. coli*, we assumed that CcrM has no drastic influence on *E. coli* physiology. As mentioned above, we now analyzed the growth rates of *E. coli* (XL1 Blue) cultures transformed with pBAD24_CcrM under expression conditions and repressed conditions. In addition, we used PBAD24_M.SssI for comparison under the same conditions. We included the results of these experiments in the Supplementary information of the article (Supplementary Fig. 16). They did not reveal strong effects of CcrM expression on viability of *E. coli*. Moreover, we now also checked growth rates of XL1-Blue cells carrying the memory system II in the on-state and in the off-state, but could not detect strong differences in the growth rates in the different states (Supplementary Fig. 17) indicating that orthogonality has been achieved at least to a certain extent.

We changed the writing of the main text in the results part now saying: [...] we set up an epigenetic memory system, aiming to use orthogonal components that do not interfere with *E. coli* physiology to consider the comment of the reviewer also at this point.

Can the authors justify why they did not investigate a system which couples methyltransferase activity to a more specific DNA binding protein, for example a zinc-finger protein or deactivated-Cas9? The programmability of other DNA binding proteins might widen the applicability of the authors' epigenetic switch principle.

Reply: Please note that the concept used here is fundamentally different from targeted methylation, which we also do in our lab. In our approach, the methyltransferase (CcrM) is expressed and expected to methylate all its GANTC target sites. Specificity of the readout is obtained by using the specific repressor protein (here a ZnF protein) with binding sites overlapping with GANTC sites. In targeted methylation, the methyltransferase is fused to a targeting device (ZnF, TAL, or CRISPR-dCAS9) and it methylates only a small subset of its recognition sites which are nearby the ZnF target sequence. For the development of the memory system, it was necessary to design the artificial repressor proteins that bind DNA in a methylation sensitive manner, and optimize all expression constructs to generate stable off- and on-states and positive feedback. This all could be achieved using a non-targeted methyltransferase, this is why we did not include this additional option.

Likewise, are zinc-finger proteins the best class of repressor to use? There is no justification given. Would TALE repressors, which are more easily programmed to specific sequences, be better? What

other classes of repressor are responsive to methylation of their binding site, and could the system be used to interface with gene circuits built from other classes of components?

Reply: TALE repressors might be also applicable as well as other DNA binding domains that are potentially sensitive to DNA methylation like helix-turn-helix motifs. We have now mentioned potential alternatives in the text. However, due to the good characterization of zinc finger proteins and the better handling in means of molecular biology compared to more repetitive TALEs, we decided to go for zinc finger proteins.

The choice of mechanism to reverse the switch also limits the expansion of this class. For example, using small RNAs to inhibit translation of the methyltransferase would allow specific enzymes to be knocked down when many are operating in parallel in a gene circuit.

Reply: Using small RNAs is a very tempting approach that also came to our mind already. One of the many fascinating options of this methodology is related to a much higher flexibility. However, we chose the mf-Ion system, because it had been described as a very robust system with very little interference. Our experiences were very positive and we can fully endorse these literature statements.

Whilst the use of DNA methylation to build a gene circuit is novel, it has been used for other bio-engineering applications. Additional references to reflect this background to the present work would be beneficial.

Other references to consider in general:

Repurposing the CRISPR-Cas9 system for targeted DNA methylation; Vojta et al; NAR; 2016

Reprogrammable CRISPR/Cas9-based system for inducing site-specific DNA methylation; McDonald et al; Biol. Open; 2016

Specific targeting of cytosine methylation to DNA sequences in vivo; Smith and Ford; NAR; 2007

Recognition of methylated DNA by TAL effectors; Deng et al; Cell Res.; 2012

Engineered zinc finger proteins that respond to DNA modification by HaeIII and HhaI methyltransferase enzymes; Isalan and Choo; JMB; 2000.

Chimeric DNA methyltransferases target DNA methylation to specific DNA sequences and repress expression of target genes; Li et al; NAR; 2007

Synthetic epigenetics—towards intelligent control of epigenetic states and cell identity; Jurkowski et al; Clin. Epigenetics; 2015

Reply: We like to reiterate that the concept used here is different from targeted methylation, and we think it would not be helpful to mix these concepts in the discussion and references provided. We very much appreciate the suggestion to cite previous examples of cytosine-C5 methylation dependent DNA binding proteins and included the Deng and Isalan citations.

Reviewers' Comments:

Reviewer #1 (Remarks to the Author)

This version of the manuscript is substantially improved but the writing needs clarification. Some of the suggested changes may be merely aesthetic but others are conceptually relevant.

Line 5, Abstract. Do you mean "experimentally validated"? Might "validated" be enough? Or simply "tested"?

Lines 14-19. This long sentence should be shortened. In the Abstract there is no need to go into details about applications.

Line 24. Delete "for various purposes".

Line 33. Delete "for the first time".

Lines 38-47. There is a wrong message here. After stating that epigenetic effects in bacteria are mediated by DNA methylation, mainly N6 adenine methylation, the authors provide examples of bacterial processes under epigenetic control. However, spore formation and biofilm formation are not known to be controlled by DNA methylation.

Lines 64-67 and 84-86. This sentence is unnecessarily duplicated.

Line 109. Choice of either British or American spelling of English language should be consistent.

Line 148. Do you mean upstream of the *ccrM* gene? Please note also that the gene name should be *ccrM*.

Line 168. Delete "the" before Znf4.

Line 170. "Demonstrate" belongs to the mathematical world. "Show" should be sufficient.

Line 177. Remove "the" before EGFP.

Line 212. Avoid "demonstrating", please.

Lines 218-220. Transmission of the on state to naive cells upon transformation is a relevant result, and should be commented and discussed in more detail.

Line 259. Do you mean "orthologous"?

Line 286. Delete "fully".

Line 321. Suggested writing: "it should be possible".

Line 328. Please explain what you mean by "DNA methylation codes".

Reviewer #3 (Remarks to the Author)

The changes to the manuscript provide useful clarification and support to the main findings; thanks to the authors for their efforts to respond to my initial points.

I still have doubts about the significance of this work. The idea of using a methyltransferase as a biological bistable switch is novel (though the principle and applications of bistable switches are well established), and the data convince me that the system works as described. However, to be a useful addition to the synthetic biology toolkit, the extent to which this methylation system interacts with native cellular regulation and other synthetic components must be clearly defined (and preferably minimal with regard to the host metabolism). Useful tool families also contain many members that can act in parallel to form more complex control networks. Whilst theoretically possible for this type of synthetic regulation, it has not been demonstrated here, and the large number of target sites in the genome makes me concerned about how many methyltransferases could operate in parallel without significantly burdening the host organism.

The new Supplementary Figure 1 shows that, though smaller than that of M.SssI, there is a definite impact of CcrM expression on growth rate. To what extent does this burden render the population of cells harbouring the switch vulnerable to escape mutants? And does the burden arise through CcrM expression causing mis-regulation of other genes? This is an important consideration if CcrM causes mis-regulation of metabolic genes crucial to the effective functioning of an engineered strain. Orthogonality is a key principle of synthetic biology; it underpins the predictability and reliability of an engineered system.

The authors state that they do not aim to compare their system to existing bistable switches, instead aiming to demonstrate feasibility (which they clearly do). There is no problem with that, but it must be accepted that doing so limits the impact of the manuscript: how do other researchers know if this type of bistable switch might be appropriate to use in their system?

There is no discussion about which other methyltransferases might be used in this system. Can it be predicted which might function well, and which might be toxic? I know it would take time and resources to demonstrate the system working with a different methyltransferase, but if the switch is not easily adaptable then surely this is a barrier to scale-up that ought to be acknowledged?

I understand that it is probably beyond the scope of the present study to respond to these issues, but in that case a frank discussion of the limitations of the system is required, setting out future issues that need to be addressed if this switch is to prove as useful as one might hope. The work presented is solid, but I do not feel it has demonstrated high enough significance for publication in this journal.

In case they are required, here are a few more specific points for improving the manuscript.

There is no data as to why ZnF4 was used in preference over ZnF2; including ZnF4 in the Supp Fig 6 comparison would be good. I think this is important because it influences the design requirements of the operator sequence which is useful information for those wishing to re-use the system.

There is no analysis of significance in any of the figures, and some experiments are only performed in duplicate.

Combine replicates in Supp Figs 14+15, add statistics, plot as a scatter graph of fluorescence vs concentration or time. Use the data to identify the threshold inducer concentration and exposure period to produce stable switching.

Manuscript NCOMMS-16-26741A

Reviewer #1 (Remarks to the Author)

This version of the manuscript is substantially improved but the writing needs clarification. Some of the suggested changes may be merely aesthetic but others are conceptually relevant.

Reply: Thanks a lot for the positive assessment and your very valuable further input.

Line 5, Abstract. Do you mean "experimentally validated"? Might "validated" be enough? Or simply "tested"?

Reply: This has been changed to "experimentally validated".

Lines 14-19. This long sentence should be shortened. In the Abstract there is no need to go into details about applications.

Reply: The sentence has been shortened and rewritten.

Line 24. Delete "for various purposes".

Line 33. Delete "for the first time".

Reply: Both corrected.

Lines 38-47. There is a wrong message here. After stating that epigenetic effects in bacteria are mediated by DNA methylation, mainly N6 adenine methylation, the authors provide examples of bacterial processes under epigenetic control. However, spore formation and biofilm formation are not known to be controlled by DNA methylation.

Reply: We have rewritten the corresponding paragraph in the introduction. We added phage protection by RM system as an important biological function of bacterial DNA methylation some lines above.

Lines 64-67 and 84-86. This sentence is unnecessarily duplicated.

Reply: We have rewritten the sentence at its second occurrence. We think the information should be kept at both places to ensure the proper flow of the text.

Line 109. Choice of either British or American spelling of English language should be consistent.

Line 148. Do you mean upstream of the *ccrM* gene? Please note also that the gene name should be *ccrM*.

Line 168. Delete "the" before ZnF4.

Reply: All corrected.

Line 170. "Demonstrate" belongs to the mathematical world. "Show" should be sufficient.

Reply: We have changed the verb to "confirm".

Line 177. Remove "the" before EGFP.

Reply: Done.

Line 212. Avoid "demonstrating", please.

Reply: It has been changed to "showing".

Lines 218-220. Transmission of the on state to naive cells upon transformation is a relevant result, and should be commented and discussed in more detail.

Reply: We have added one sentence to the discussion highlighting this point a bit more "The methylation signal that encodes the on-state of the system was stable over many cell generations and during physiological changes including the generation of competent cells and transformation of bacteria."

Line 259. Do you mean "orthologous"?

Line 286. Delete "fully".

Line 321. Suggested writing: "it should be possible".

Reply: All corrected.

Line 328. Please explain what you mean by "DNA methylation codes".

Reply: We have replaced the term "codes" by "patterns" and expanded this section.

Reviewer #3 (Remarks to the Author)

The changes to the manuscript provide useful clarification and support to the main findings; thanks to the authors for their efforts to respond to my initial points.

Reply: Thank you for these positive words. We also appreciate your further input very much and tried to accommodate as much of it as possible.

I still have doubts about the significance of this work. The idea of using a methyltransferase as a biological bistable switch is novel (though the principle and applications of bistable switches are well established), and the data convince me that the system works as described. However, to be a useful addition to the synthetic biology toolkit, the extent to which this methylation system interacts with native cellular regulation and other synthetic components must be clearly defined (and preferably minimal with regard to the host metabolism). Useful tool families also contain many members that can act in parallel to form more complex control networks. Whilst theoretically possible for this type of synthetic regulation, it has not been demonstrated here, and the large number of target sites in the genome makes me concerned about how many methyltransferases could operate in parallel without significantly burdening the host organism.

Reply: Many years of work with bacterial DNA MTases has provided numerous examples of enzymes from this group that can be expressed in other bacteria without strong physiological consequences. Perhaps the best examples are the EcoRI and EcoRV MTases, which occur in natural E. coli isolates and can be expressed in lab strains without big difficulties. This information has been added to the discussion of our manuscript.

The new Supplementary Figure 1 shows that, though smaller than that of M.SssI, there is a definite impact of CcrM expression on growth rate. To what extent does this burden render the population of cells harbouring the switch vulnerable to escape mutants? And does the burden arise through CcrM expression causing mis-regulation of other genes? This is an important consideration if CcrM causes mis-regulation of metabolic genes crucial to the effective functioning of an engineered strain. Orthogonality is a key principle of synthetic biology; it underpins the predictability and reliability of an engineered system.

Reply: First of all, we like to mention, that no effect on viability was observed after 3 hours of expression and the effect after 5 hours was rather small.

We have conducted additional growth studies using an empty NusA-His₆ tag pBAD construct to estimate the unspecific effect of pBAD induction on E. coli growth. Moreover, we also used the catalytically inactive CcrM mutant. Both experiments showed that there is no CcrM activity dependent effect of CcrM expression on viability of E. coli, which answers the question of the reviewer. This information has now been added to the manuscript in the results section and in Suppl. Fig. 1B. The small reduction of growth rates observed after 5 hours of CcrM expression is related to the general overexpression conditions in the pBAD system. Please note that these initial viability tests were conducted under conditions of CcrM overexpression, while the memory system uses the maintenance CcrM, which is expressed at lower levels. These data are now presented in a new chapter in the results section.

Additionally, we have inspected the core binding sites of all E. coli transcriptional regulators as listed in the regulon database, which contains 3436 entries. Among them only 127 sites contain a GANTC

sequence (3.6%). At this point, one further needs to consider that only a fraction of genes are needed under experimental growth conditions, and that the presence of a methylation site in a TF binding sites does not automatically imply that methylation indeed influences DNA binding. Based on this estimation, one would expect minimal effects of CcrM methylation in *E. coli*, which is in line with our experimental results of a minimal physiological effect. This information has been added to the manuscript in Suppl. Fig. 1.

The authors state that they do not aim to compare their system to existing bistable switches, instead aiming to demonstrate feasibility (which they clearly do). There is no problem with that, but it must be accepted that doing so limits the impact of the manuscript: how do other researchers know if this type of bistable switch might be appropriate to use in their system?

Reply: We have added sentences in the discussion section describing future work and limitations of the current study, in which we state, that studies as proposed by the reviewer need to be done. We believe it will be necessary to focus on particular applications and based on the applicative needs select the most suitable system. Moreover, we have added two more literature citations to our list of existing genetic switch systems. This illustrates a large variety of design principles, making it very difficult to compare all of them directly with our system.

There is no discussion about which other methyltransferases might be used in this system. Can it be predicted which might function well, and which might be toxic? I know it would take time and resources to demonstrate the system working with a different methyltransferase, but if the switch is not easily adaptable then surely this is a barrier to scale-up that ought to be acknowledged?

Reply: Please refer to our response above. There is strong evidence that M.EcoRV or M.EcoRI will work without problems. This is now mentioned in the discussion section.

I understand that it is probably beyond the scope of the present study to respond to these issues, but in that case a frank discussion of the limitations of the system is required, setting out future issues that need to be addressed if this switch is to prove as useful as one might hope. The work presented is solid, but I do not feel it has demonstrated high enough significance for publication in this journal.

Reply: We have included statements about additional work necessary in the discussion. This includes A) the proposal to measure all critical concentrations necessary for modelling of the system, B) the request to compare properties of our system with existing system including sensitivity, stability, and reversibility in the light of the requirement of the different applications.

We like to mention, that our system is the first example of a working DNA methylation based epigenetic toggle system which combines the epigenetic key features of stability and reversibility. We believe that this represents a high value by itself, beyond the potential immediate biotechnological application.

In case they are required, here are a few more specific points for improving the manuscript.

There is no data as to why ZnF4 was used in preference over ZnF2; including ZnF4 in the Supp Fig 6 comparison would be good. I think this is important because it influences the design requirements of the operator sequence which is useful information for those wishing to re-use the system.

Reply: ZnF2 was an intermediate in the design process. After obtaining ZnF4, all experiments were started using ZnF4, the putatively more potent factor. Since ZnF4 turned out to be fully functional, we did not return to ZnF2.

There is no analysis of significance in any of the figures, and some experiments are only performed in duplicate.

Reply: Thanks for this hint. We have shown all data with error bars, but we now provide in Suppl. Table 1 all p-values for the bar diagrams in the main article as well.

Combine replicates in Supp Figs 14+15, add statistics, plot as a scatter graph of fluorescence vs concentration or time. Use the data to identify the threshold inducer concentration and exposure period to produce stable switching.

Reply: As mentioned in our original response to your comments, a detailed analysis of the switching behavior of our system goes far beyond the scope of this paper. Our data show a complicated interdependence of inducer concentrations and inductions times indicating that a two dimensional grid of variations in both parameters needs to be explored. Moreover, in our view such analysis should be accompanied by a quantitative analysis of the concentrations of all critical factors (mainly trigger and maintenance CcrM and ZnF, in each case protein and mRNA), and methylation rates of operator regions after replication. All these parameters will be needed to model the system allowing to predict its behavior and rationally improve it.